# Lifelong Exposure to a Low-Dose of the Glyphosate-Based Herbicide RoundUp^®^ Causes Intestinal Damage, Gut Dysbiosis, and Behavioral Changes in Mice

**DOI:** 10.3390/ijms23105583

**Published:** 2022-05-17

**Authors:** Ingrid Del Castilo, Arthur S. Neumann, Felipe S. Lemos, Marco A. De Bastiani, Felipe L. Oliveira, Eduardo R. Zimmer, Amanda M. Rêgo, Cristiane C. P. Hardoim, Luis Caetano M. Antunes, Flávio A. Lara, Claudia P. Figueiredo, Julia R. Clarke

**Affiliations:** 1Faculdade de Farmácia, Universidade Federal do Rio de Janeiro, Rio de Janeiro 21941-902, RJ, Brazil; delcastiloingrid@gmail.com (I.D.C.); claufig@gmail.com (C.P.F.); 2Instituto de Ciências Biomédicas, Universidade Federal do Rio de Janeiro, Rio de Janeiro 21941-902, RJ, Brazil; arthurneumann19@gmail.com (A.S.N.); simoeslemos.f@gmail.com (F.S.L.); felipe@histo.ufrj.br (F.L.O.); 3Departamento de Farmacologia, Universidade Federal do Rio Grande do Sul, Porto Alegre 90040-193, RS, Brazil; tyrev@hotmail.com (M.A.D.B.); erzimmer@gmail.com (E.R.Z.); 4Instituto Oswaldo Cruz, Fundação Oswaldo Cruz, Rio de Janeiro 21040-360, RJ, Brazil; amanda.biorj@gmail.com (A.M.R.); caetano.antunes@fiocruz.br (L.C.M.A.); flavioalveslara2000@gmail.com (F.A.L.); 5Instituto de Biociências, Universidade Estadual Paulista, São Vicente 11380-972, SP, Brazil; cristianehardoim@gmail.com; 6Instituto Nacional de Ciência e Tecnologia de Inovação em Doenças de Populações Negligenciadas, Centro de Desenvolvimento Tecnológico em Saúde, Fundação Oswaldo Cruz, Rio de Janeiro 21040-361, RJ, Brazil

**Keywords:** herbicide, repetitive behavior, social impairment, neurodevelopmental diseases, gut–brain axis, inflammation

## Abstract

RoundUp^®^ (RUp) is a comercial formulation containing glyphosate (N-(phosphono-methyl) glycine), and is the world’s leading wide-spectrum herbicide used in agriculture. Supporters of the broad use of glyphosate-based herbicides (GBH) claim they are innocuous to humans, since the active compound acts on the inhibition of enzymes which are absent in human cells. However, the neurotoxic effects of GBH have already been shown in many animal models. Further, these formulations were shown to disrupt the microbiome of different species. Here, we investigated the effects of a lifelong exposure to low doses of the GBH-RUp on the gut environment, including morphological and microbiome changes. We also aimed to determine whether exposure to GBH-RUp could harm the developing brain and lead to behavioral changes in adult mice. To this end, animals were exposed to GBH-RUp in drinking water from pregnancy to adulthood. GBH-RUp-exposed mice had no changes in cognitive function, but developed impaired social behavior and increased repetitive behavior. GBH-Rup-exposed mice also showed an activation of phagocytic cells (Iba-1–positive) in the cortical brain tissue. GBH-RUp exposure caused increased mucus production and the infiltration of plama cells (CD138-positive), with a reduction in phagocytic cells. Long-term exposure to GBH-RUp also induced changes in intestinal integrity, as demonstrated by the altered expression of tight junction effector proteins (*ZO-1* and *ZO-2*) and a change in the distribution of syndecan-1 proteoglycan. The herbicide also led to changes in the gut microbiome composition, which is also crucial for the establishment of the intestinal barrier. Altogether, our findings suggest that long-term GBH-RUp exposure leads to morphological and functional changes in the gut, which correlate with behavioral changes that are similar to those observed in patients with neurodevelopmental disorders.

## 1. Introduction

Glyphosate (N-(phosphono-methyl) glycine) is the world’s leading wide-spectrum herbicide used in agriculture. Glyphosate-based herbicides (GBH), including RoundUp^®^ (RUp), are a mixture of glyphosate and other undisclosed additives that enhance their effect [1]. Studies have shown that glyphosate is a contaminant of food and water consumed by humans [2]. Experimental evidence has shown that glyphosate can cross the intestinal–epithelial barrier [3]. Glyphosate was also shown to be absorbed by humans and excreted in feces and urine, where its concentration was shown to vary between 1 and 10 µg/L [4]. Supporters of the broad use of GBH claim it is innocuous to humans, since it is an inhibitor of the shikimate pathway, which is absent in human cells [5,6]. However, the safety and toxicity of GBH are a matter of extensive debate in the scientific community [5,6,7].

The neurotoxic effects of GBH have already been shown in many animal models. Experimental evidence has shown that GBHs induce oxidative stress, apoptosis, and disbalance in neurotransmitter systems [8,9,10,11]. Gallegos and colleagues have demonstrated that RUp ingestion during pregnancy and lactation in rats was also shown to impact the structure and function of the brains of the offspring [12]. However, the molecular mechanisms behind these neural injuries remain unknown.

The gut microbiota consists of the collection of microbes that are present within the human gastrointestinal tract (GIT) [13], and its imbalance (dysbiosis) has been implicated as a potential cause of several diseases [14]. The brain, in particular, was shown to be especially sensitive to molecules produced by the bacteria colonizing the GIT [15,16]. Alterations in GIT microbiome are related with neuropsychiatric and neurodegenerative disorders, especially autism. One of the proposed mechanisms of the gut–brain axis involves changes in intestinal permeability, allowing bacteria, toxic metabolites, and small molecules to reach the bloodstream and, possibly, the nervous system. Increasing evidence shows that GBHs interfere with the growth of bacteria that typically colonize the human gut [17]. One recent study described that a sub-chronic exposure to high doses of glyphosate results in morphological alterations in the gut of female rats [18]. However, the long-term effects of low-doses of GBH remain largely unknown.

Here, we investigated the effects of lifelong exposure to low doses of the GBH RoundUp^®^ (GBH-RUp) on gut environment, including morphological and microbiome changes. We also aimed to determine whether exposure to GBH-RUp could harm the developing brain and lead to long-term behavioral changes in adulthood. To this end, mice were exposed to GBH-RUp in drinking water from pregnancy to adulthood. GBH-exposed mice developed impaired social behavior and increased repetitive behavior. The intestine of GBH-RUp-exposed mice showed a larger area occupied by mucus-producing cells, and changes in leukocyte population. This was accompanied by changes in the GIT microbiome composition and the expression of tight junction effector proteins, which are crucial for the establishment of the intestinal barrier. Altogether, our findings suggest that long-term GBH-RUp exposure leads to morphological and functional changes in the gut, which are associated with the development of behavioral changes similar to those observed in patients with neurodevelopmental disorders.

## 2. Results

In order to investigate the behavioral effects of lifelong exposure to low amounts of glyphosate, mice were treated with GBH-RUp in drinking water (0.075% *w/v*) from pregnancy until adulthood. Mice were initially tested in the open field (OF) task, which measures locomotor and anxiety behaviors. No differences were seen in the distance traveled (data not shown) and in the time spent by male and female mice in the center of the arena (Figure 1A,B), suggesting that GBH-RUp exposure does not induce changes in locomotor and anxiety behaviors. We then evaluated whether long-term GBH-RUp exposure interfered with normal memory formation. We found that both control and GBH-Rup-exposed mice showed similar performance in the novel object recognition task (NOR), as shown by the similar recognition index for both groups among male (Figure 1C) and female (Figure 1D) mice. To further investigate the effects of GBH-RUp on memory formation, control and GBH-RUp-exposed mice were trained in the Morris Water Maze (MWM) paradigm. Both control and treated mice showed a significant decrease in the time to find the submerged platform (escape latency) when comparing the first and last days of training, suggesting that GBH-RUp exposure had no effect on spatial memory acquisition (Figure 1E,F). Exposure to pesticides and herbicides have already been associated with an increased risk of neuropsychiatric disorder development [19,20], including autism and schizophrenia [21,22]. We trained mice in the social approach test (SA) and found that male control mice explored the compartment containing a stranger mouse for longer periods than the empty compartment (Figure 1G), demonstrating normal sociability. Remarkably, GBH-RUp-exposed male animals showed decreased social interest, as demonstrated by the reduced time spent exploring the stranger mouse (Figure 1G). In contrast, no difference in social interest was seen between female GBH-RUp-exposed mice and the control group, since both groups exhibited comparable exploration time towards the stranger mouse and the empty compartment (Figure 1H). Repetitive digging of marbles in the marble-burying (MB) task is frequently used to assess stereotyped behavior in several mouse models of autism. We found that both male (Figure 1I) and female (Figure 1J) GBH-RUp-exposed animals buried significantly more marbles than control mice, indicating increased repetitive behavior. Altogether, these findings suggest that long-term GBH-RUp exposure leads to behavioral alterations, some of which were sex-specific.

We next investigated whether behavioral alterations were associated with gliosis in the brains of mice following long-term exposure to GBH-RUp. Brain sections obtained from adult mice were immunostained for glial fibrillary acidic protein (GFAP), an astrocyte marker, and ionized calcium binding adaptor molecule 1 (Iba-1, a macrophage/microglial marker). No differences in GFAP (Figure 2A–C) or Iba-1 (Figure 2D–F) immunoreactivity were detected in the cortex of GBH-RUp-exposed mice when compared to control animals. In contrast, we found that phagocytic cells were activated in the cortex of GBH-RUp-exposed mice, as demonstrated by the reduced Feret diameter (Figure 2G) and the increased Iba-1 immunoreactivity per cell (Figure 2H). In addition, long-term GBH-RUp exposure was not associated with increased cell death in the cortex of mice (data not shown). Altogether, these findings suggest that long-term GBH-RUp exposure is associated with morphological changes in brain-resident glial cells.

Mounting evidence suggests that changes in gut structure and integrity as well as in the gut microbiome can influence brain functions [23,24,25]. Accordingly, we decided to investigate whether long-term exposure to GBH-RUp induced changes in intestinal integrity and function. Goblet cells can largely influence the innate defense system by producing mucus [26]. We found that colonic goblet cells had an increased area of PAS (periodic acidic-Schiff) staining (Figure 3A–C), whereas the number of cells per crypt was comparable between groups (Figure 3D), suggesting that larger amounts of mucus are produced in GBH-RUp-exposed animals. We also investigated whether GBH-RUp exposure affected gut leukocyte population and found a decreased number of Iba-1-positive cells in the gut lamina propria of GBH-RUp mice (Figure 3E–G). A significant increase in Arginase-1 immunostaining was also observed in the gut of GBH-RUp exposed mice when compared to control animals (Figure 3H–J), indicating the predominance of alternatively activated macrophages in colonic tissue of mice who were exposed to the herbicide. H&E staining in the gut lamina propria revealed that both control and GBH-RUp-exposed mice had a significant number of cells that were morphologically compatible with lymphocytes (rounded cells with high nucleus/cytoplasm relation) and plasma cells (oval basophilic cells with eccentric nucleus and perinuclear halo containing Golgi apparatus) (Figure 3K–M). Then, we evaluated colon CD8- and CD-138-positive lymphocytes, markers of CD8 lymphocytes and plasma cells, respectively. We found that both CD8 (Figure 3N–P) and CD138 (Figure 3Q–S) leukocyte populations were significantly increased in GBH-RUp-exposed mice.

To further classify the clusters of cells observed in the intestinal lamina propria of both control and GBH-RUp-exposed mice, colon samples were immunostained with anti-B220 (B lymphocytes) and anti-Blimp-1 (Ig-secreting cells) antibodies. We found that B220^+^ cell niches were distributed throughout the lamina propria (Figure 4A), whereas in GBH-RUp-exposed mice, B220^+^ cells were frequently spread throughout the lamina propria, outside of these clusters (Figure 4B, white arrowheads). Regarding Blimp-1^+^ plasma cells, they were poorly observed in B cell-enriched regions in both control and GBH-RUp-exposed mice (Figure 4C,D). However, a significant number of non-clustered B cells coexpressed both lymphocyte markers (B200^+^-Blimp-1^+^ cells) in colon samples from GBH-RUp-exposed mice (Figure 4E,F, white arrowheads). Given that Ki67 immunostaining did not differ between the control and GBH-RUp-exposed mice (Figure 4G–I), these data suggest that herbicide exposure induced the differentiation of B lymphocytes into plasma cells in the colon of mice.

Changes in the gut’s structure often lead to complete or partial loss of the integrity of the intestinal barrier [27], increasing the circulation of toxins, toxic metabolites, and even bacteria [28,29]. We then investigated whether GBH-RUp exposure interferes with the expression of important effectors of intestinal tight junctions, proteins that mediate cell–cell interactions and maintain the homeostasis of the intestinal barrier. mRNA levels of *claudin-4* (*CLDN-4*) were comparable between groups when evaluated in 30 (Figure 5A) and 60 day-old animals (Figure 5B). In contrast, we found a decreasing trend in the expression of *zonula occludens* (*ZO*)-*2* in the gut from 30 day-old GBH-RUp-exposed animals when compared to age-matched controls (Figure 5C), and an opposite effect when evaluated in 60 day-old animals, with an increased expression of *ZO-2* demonstrated in GBH-RUp mice (Figure 5D). At the protein level, we performed immunofluorescence targeting the zonula occludens (ZO)-1 tight junction protein, with no significant difference being detected between both experimental group in 60 day-old mice (Figure 5E–G). We further evaluated the expression of syndecan-1, a proteoglycan that contributes to the homeostasis of intestinal barrier [30]. Our data demonstrated that the distribution of syndecan-1 was significantly different in the gut of GBH-RUp-exposed mice when compared to control animals, changing from the basal surface of the gut epithelium to becoming concentrated in a more lateral/apical location (Figure 5H–I”). Taken together, our findings suggest that the long-term exposure to low amounts of GBH leads to significant changes in the intestinal mucosa structure, which might account for the behavioral changes seen in our model.

Next, we examined whether long-term GBH-RUp exposure was associated with changes in gut bacterial composition. No significant changes in alpha diversity were observed when the microbiota composition of control and GBH-RUp-exposed mice were compared (females, *p* = 0.4944; males, *p* = 0.1809; Figure 6A). However, the relative abundances of some phyla were significantly altered by GBH-RUp exposure (Figure 6B). Of relevance, GBH-RUp exposure caused a significant increase in the relative abundance of *Proteobacteria* and *Desulfobacteria* phyla in female mice when compared to controls (Figure 6C). Interestingly, this effect was sex specific, as these phyla did not have their relative levels significantly altered in the male mice. Instead, in the male mice, a modest but statistically significant decrease in the relative abundance of taxa belonging to the *Bacteroideta phylum* (Figure 6D) was observed in GBH-RUp-exposed mice when compared to controls. Next, we used LEfSe to determine the top discriminating bacterial taxa (species-level) when GBH-RUp-treated and -untreated animals were compared. Figure 6 shows the top 10 discriminating taxa in male (Figure 6E) and female (Figure 6F) mice. Although several species were found, it is worth noting that 6/10 species in the female and 5/10 species in the male mice were members of the *Lachnospiraceae* family.

## 3. Discussion

The developing brain is extremely sensitive to the impact of environmental adversities. Perinatal infections, as well as drug and alcohol exposure, among others, are factors known to increase the risk of development of neurodevelopmental diseases, including autism [31,32]. Glyphosate-based formulations are among the most widely used herbicides in agriculture worldwide [2,33,34]. Exposure to pesticides and herbicides have already been associated with autism [32,34,35,36], but a causal association and the mechanisms linking these two factors remain to be established.

The neurotoxic effects of GBH were already shown under different conditions, in both patients and animal models. Delirium, confusion, anxiety, and short-term memory impairments were described in the case reports of patients who had ingested large amounts of glyphosate-based herbicides when they attempted suicide [37,38]. In rodents and fish, an acute exposure to GBH was associated with memory impairment and reduced acetylcholinesterase activity [39,40]. The short-term treatment of mice with high doses of glyphosate also led to the downregulation of dopamine and serotonin markers in the brain [41]. In addition, increased anxiety levels and impaired short-term memory formation were seen in mice subjected to intranasal administration of GBH [42], suggesting that deleterious effects on brain functions may occur even without oral ingestion of glyphosate, which is especially concerning for agricultural workers. Of particular interest, one previous study reported impaired sociability in mice who were exposed to higher doses of glyphosate during pregnancy and lactation [12,43]. However, few studies have addressed the neurological and behavioral effects of chronic exposure to low doses of GBH, which better mimic the human condition. Here, we found that a long-term exposure to glyphosate-based RoundUp^®^ formulation was associated with increased repetitive behavior and impaired social interest. Some of these effects were sex-specific, with an effect of GBH-RUp only on male mice. This is in agreement with the distribution of neuropsychiatric disorders within the general population, where male children are most commonly affected. In addition, under our conditions, long-term exposure to the herbicide was not associated with changes in anxiety levels or memory impairment, which might be due to the fact that lower doses of GBH-RUp were used in our study when compared to others.

Several studies suggest that glial cells can be programed by perinatal environmental changes, contributing to late behavioral consequences [44,45,46]. Acute perinatal exposure to herbicides and pesticides was shown to induce gliosis in different animal models [47,48]. Previous studies involving acute or sub chronic exposure to high doses of GBH have shown morphological alterations in both microglial cells and astrocytes in juvenile mice [49]. Here, we report that long-term exposure to GBH-RUp, from the neonatal period into adulthood, leads to changes in microglial cell morphology. Further experiments are required in order to address whether these cells undergo epigenetic changes during development in animals exposed to GBH-RUp, and whether these alterations are partially or completely linked to the behavioral changes described.

Emerging data on the gut–microbiota–brain connection suggest that there is a link between changes in the gut microbiome composition and the development of neurological diseases [23,25,50,51]. This is especially the case for autism spectrum disorder, where data from patients and experimental models have demonstrated dysbiosis, increased intestinal permeability, and endotoxin translocation [16,24,52,53]. Glyphosate and GBH were shown to disrupt the microbiome of different species [54,55,56]. Our results are in agreement with these previous studies, since significant changes in microbiota composition were observed even at the phyla level, with relative levels of *Proteobacteria*, *Desulfobacterota*, and *Bacteroideta* being altered in GBH-Rup-exposed mice when compared to controls. In addition to these changes in the relative levels of some microbiota bacterial phyla, we found that GBH-RUp induced significant morphological changes in the gut of mice. Among these alterations, we found an increased area of mucus-producing cells, and an increased differentiation of lymphocytes into plasma cells. The altered expression of several adhesion molecules is suggestive of an altered gut permeability, or a compensatory response. Previous studies have shown that the levels of these cell–cell adhesion molecules are reduced in mouse models of autistic-like behavior [57], and fecal transplantation from an autistic donor into healthy mice led to increased gut permeability and the decreased expression of *ZO-2* [58]. Our results suggest that GBH can directly affect B cell maturation/differentiation into plasma cells, which correlates with the increased risk of multiple myeloma and lymphoma, which was previously reported following glyphosate exposure [59]. Altogether, these findings comprise the first evidence that long-term exposure to low doses of GBH-RUp leads to significant changes in the structure of the intestinal mucosa. We hypothesize that changes in the expression of adhesion molecules and in leukocyte profiles of the gut is triggered by the direct contact of glyphosate with the intestinal barrier. This would lead to increased gut permeability, ultimately leading to the behavioral alterations seen in adult animals exposed to GBH-RUp. However, the extent to which these intestinal modifications contribute to changes in brain and behavior in our model remains to be established.

It is increasingly recognized that both genetic and environmental factors influence, to different extents, the onset of neurodevelopmental disorders such as autism [60]. Under this premise, it is reasonable to assume that GBH-RUp acts as one contributing factor to increase the risk of autism development, adding to genetic and other environmental factors. The individual and cumulative weights of each of the genetic and environmental pre- and post-natal factors remains unknown and might change between subjects. However, our results provide clear evidence that the long-term exposure to low amounts of GBH-RUp causes extensive impact to both the brain and the gut, suggesting that the widespread and generalized use of GBH should be reconsidered by health authorities in different countries.

One limitation of our study involves the fact that only GBH-RUp formulation was used, instead of pure glyphosate. Commercially available formulations contain auxiliary agents and surfactants which are often undisclosed, and are supposed to enhance the active ingredient’s stability and plant penetrance. In vitro studies have already suggested that GBHs are frequently more toxic than isolated glyphosate itself [61]. However, agricultural practices are based on glyphosate-based formulations, with RUp being the most frequently used worldwide [36,61]. For this reason, the results we describe cannot be attributed solely to glyphosate, but could be associated with other components of the formulation or even to the interaction between different ingredients in the formulation. More transparency should be required from manufacturers concerning the true composition of GBHs. Indeed, further studies should address whether long-term exposure to glyphosate alone causes similar effects to the gut, brain, and behavior of rodents as described in our study. Finally, another important caveat is that the selected dose used in our study was higher than the expected daily exposure of the general population [62], although it still represents a low dose when compared to other studies performed in rodents [41].

In conclusion, our study showed that GBH-RUp is harmful to the gut microbiome of mice and causes significant changes in the gut mucosa. These effects are associated with late deleterious effects on the mouse brain and behavior. Altogether, these findings reinforce the idea that GBH-RUp may not be innocuous to human beings and that the widespread use of this herbicide should be a matter of concern for health and agricultural authorities worldwide.

## 4. Materials and Methods

Animals and Glyphosate exposure. All procedures used in the present study followed the “Principles of Laboratory Animal Care” (US National Institutes of Health) and the ARRIVE guidelines, and were approved by the Institutional Animal Care and Use Committee of the Federal University of Rio de Janeiro (protocol #163/2018). Naïve 10 week-old female Swiss mice were obtained from our facilities and were mated for two weeks (four females per male). After this period, dams were housed individually until delivery. Glyphosate-based herbicide (commercially available as Round Up^®^, GBH-RUp) was added to the drinking water of pregnant females, at a final concentration of 0.075% w/v. Animals were exposed to GBH-RUp from pregnancy, through lactation, weaning, and until adulthood. Behavioral tests were performed between postnatal days 60 (P60) and 80 (P80). Control groups received regular drinking water during this period. GBH-RUp in drinking bottles was replaced by a fresh solution every three days. No difference in palatability was detected, since mice from both groups drank similar amounts of water throughout the experiment (data not shown). From P7, pups were evaluated daily for body weight and any sign of dehydration. All litters were normalized to a maximum of 10 pups. Mice were weaned at P21, were housed with same-sex littermates (2–5 mice per cage) in polypropylene cages maintained at 25 °C with controlled humidity, under a 12 h light/dark cycle, and with *ad libitum* access to water and chow. No more than two pups per dam were used for behavioral assessments.

Open field test. Mice were individually placed at the center of an arena (30 × 30 × 45 cm). Time spent in the center of the apparatus was recorded for 5 min by ANY-maze software (Stoelting Company, Wood Dale, IL, USA) as a measure of anxiety-like behavior [63]. The arena was thoroughly cleaned with 70% ethanol between trials to eliminate olfactory cues and illuminated with an indirect source of light (~100 lux).

Novel object recognition test. The test was carried out on the same day and in the same box used for the open field test. Test objects were made of plastic and had different shapes, colors, sizes, and textures. During behavioral sessions, objects were fixed to the box using tape to prevent displacement caused by the exploratory activity of the animals. Preliminary tests showed that none of the objects used in the experiments evoked innate preference. Before training, each animal was submitted to a 5 min-long habituation session, in which it was allowed to freely explore the empty arena. Training consisted of a 5 min-long session during which animals were placed at the center of the arena in the presence of two identical objects. The time spent exploring each object was recorded by a trained researcher. Sniffing and touching of the object were considered exploratory behaviors. The arena and objects were cleaned thoroughly with 70% ethanol between trials to eliminate olfactory cues. Ninety minutes after training, animals were placed in the arena again for the test session, in which one of the objects that was used in the training session was replaced by a new one. Again, the time spent exploring familiar and novel objects was measured. All animals showed comparable exploration time towards the objects used in the training and test sessions. The recognition index was calculated by subtracting the time exploring the familiar object from the time exploring the novel object, divided by the total exploration time. Animals that successfully learn the task explore the novel object longer than the familiar object, thus showing a recognition index > 0 [64].

Social approach test. The social approach test was performed in a three-chamber apparatus, which consisted of a rectangular, transparent, acrylic box (60 × 45 × 30 cm), with two walls dividing the box in three equal chambers of 20 × 45 × 30 cm each. Cylindrical aluminum cages of 8 cm in diameter (9.5 cm height) were used to contain a stranger mouse (a mouse with which the test mice had never had any contact before). Sociability was assessed by placing one empty cage in one lateral chamber and one cage containing the stranger mouse in the opposite chamber [65,66]. The test mouse was kept in the middle chamber for 5 min, after which the walls were removed, and the mouse was allowed to fully explore the apparatus for 10 min. The time exploring each cage was manually evaluated by a trained researcher who was blind to the experimental condition. Social preference index was calculated by subtracting the time spent exploring the empty cage from the time spent exploring the stranger mouse, divided by the total exploration time.

Marble burying test. The experiment was performed in standard polysulfone house cages (32 × 20 × 21 cm) with fresh, unscented bedding to a depth of 4 cm, which was hard-pressed until forming an even surface. During the habituation phase, mice were individually placed in the cage and were allowed to freely explore for 10 min. During the test phase, 20 dark glass marbles (11 mm diameter) were gently placed equidistant in a 4 × 5 arrangement and after 20 min, the number of marbles which had more than 2/3 of their area covered with bedding was counted [65,67].

Morris Water Maze. This test was performed in a round pool measuring 58 cm diameter × 122 cm high, which was filled with water and placed in a room with spatial clues on all walls. Water temperature was kept at 24 ± 2 °C. Training sessions consisted of four trials per day during four consecutive days and, in each trial, mice were placed in the pool from a different starting location. A platform with a diameter of 10.5 cm was placed on a fixed location and submerged 1 cm below the water level. The time taken for animals to reach the platform (escape latency) in each trial was counted by an experienced researcher. If animals took more than 60 s to find the hidden platform, they were removed from the water, placed on top of the platform and allowed to stand there for 30 s [68]).

Tissue preparation and morphological analysis. Mice were deeply anesthetized with ketamine (80 mg/kg) and xylazine (10 mg/kg) and colon samples were dissected, washed in sterile phosphate-buffered saline (PBS), and immediately immersed in paraformaldehyde (PFA) 4% for 48 h. For brain collection, animals were perfused transcardially with 50 mL of PBS (0.1 M, pH 7.4) per animal followed by ice-cold 4% PFA. After fixation, tissues were included in paraffin blocks and 5 µm-thick slices were obtained in a Leica microtome (Leica, Wetzlar, Germany).

For general histological analysis, colonic sections were stained with hematoxylin and eosin (H&E) and were imaged by light microscopy. For Periodic Acid-Schiff (PAS) staining, sections were washed, deparaffinized, rehydrated, and immersed in 0.5% periodic acid solution for 5 min. Sections were then washed in distilled water and immersed in Schiff reagent for 15 min, and counterstained with Hematoxylin. Images were obtained in an Eclipse 50i light microscope (Nikon, Tokyo, Japan).

For immunofluorescence, paraffin-embedded colonic and cerebral samples were immersed in xylene for 10 min, rehydrated by incubation in absolute ethanol, followed by 95 and 70% solutions of ethanol in water. Slides were then heated at 95 °C in 0.01 M citrate buffer during 45 min for antigenic recovery. Primary antibodies (Table 1) were diluted in Super Block (ScyTek, Logan, UT, USA) and incubated for 12–16 h at 8 °C. After washing with PBS-T (0.25% Tween 20 in PBS), the slides were incubated with fluorescent secondary antibodies (Table 1), diluted in blocking solution for 2 h at room temperature. The slides were washed and mounted with DAPI-containing mounting medium, were covered with glass coverslips, and then imaged on a Sight DS-5M-L1 digital camera (Nikon, Melville, NY, USA) connected to an Eclipse 50i light microscope (Nikon) at different magnifications. The images acquired were used for quantifications of fluorescence intensity using Image J software (NIH) in whole tissue, as well as in individual cells. To characterize Iba-1-positive cells, we evaluated the Feret diameter of individual cells as previously described [69]. Maximum and minimum Feret are determined as the longest and most ramified diameters of a cell body, respectively. When cells are close to a perfect circle, Feret diameter tends to zero.

Colon immunohistochemistry was developed in 5 µm paraffin-embedded slices. Slides were washed for 10 min in xylene 3 times, followed by decreasing concentrations of ethanol in water (100%–90%–70%). After that, the samples were immersed in distilled water for 5 min for rehydration and were sequentially heated in citrate buffer, for antigenic recovery, for 45 min. Slides were then incubated with endogenous peroxidase-blocking solution (ScyTek) for 25 min. After washing for 15 min, primary antibodies were applied for 1 h at room temperature, and then were washed and incubated with HRP-conjugated secondary antibodies for 1 h. The peroxidase reaction was visualized with 3,3′-diaminobenzidine (DAB, Dako, CA, USA) substrate for 1 to 5 min or until a brown precipitate could be observed. Identical conditions and reaction times were used for slides from different animals (run in parallel) to allow a comparison between immunoreactivity densities. Reactions were stopped by immersion of slides in distilled water. Counter-staining was performed with Harris Hematoxylin. Slides were washed in running water, dehydrated in alcohol, cleared in xylene, mounted in resinous medium, and examined with light microscopy using a Sight DS-5M-L1 digital camera (Nikon) connected to an Eclipse 50i light microscope (Nikon) at different magnifications.

RNA extraction and qPCR analysis. Large intestine samples were collected and stored at −80 °C until RNA extraction, which was performed with Trizol (Invitrogen) according to the manufacturer’s instructions. The purity of the RNA was determined by the 260/280 and 260/230 nm absorbance ratios. Only preparations with ratios >1.8 were used. One microgram of total RNA was treated with DNAse I (ThermoFisher Scientific Inc, Waltham, MA, USA) according to manufacturer’s recommendations, before cDNA synthesis was performed using the High-Capacity cDNA Reverse Transcription Kit (ThermoFisher Scientific Inc). For gene expression analysis related to gut permeability, qPCR assays were performed using the Power SYBR kit (ThermoFisher Scientific Inc) and were run in a QuantStudio 5 (ThermoFisher Scientific Inc) to determine the cycle threshold (Ct). Delta Delta Ct was calculated, and actin was used as an endogenous control. Primer sequences are shown on Table 2. Statistical analysis was performed using the GraphPad Prism 8 program.

Microbiota composition analyses. To determine the gut microbiota bacterial composition of control and GBH-RUp-exposed animals, fresh feces were collected at post-natal day 30 and were stored at −80 °C. DNA was extracted from these samples (approximately 100 mg of feces) using the QIAamp DNA Stool Minikit (Qiagen, Hilden, Germany), with modifications. Mechanical lysis was performed in PowerBead Tubes, Garnet (Qiagen), where samples were mixed with 1 mL of InhibitEX Buffer and incubated at 95 °C for 5 min. Tubes were then placed horizontally in a vortex adapter and vortexed at the maximum speed for 10 min. Tubes were then centrifuged at 13,000× *g* for 1 min and the supernatant was transferred to a clean 2 mL tube. The remaining steps were performed according to the manufacturer’s instructions and purified DNA was stored at −80 °C until analysis.

16S rDNA sequencing was performed using standard Illumina protocols. First, variable regions V3 and V4 of the 16S rRNA gene were amplified using the primers (containing sequencing adapters) 5′-TCG TCG GCA GCG TCA GAT GTG TAT AAG AGA CAG CCT ACG GGN GGC WGC AG-3′ and 5′-GTC TCG TGG GCT CGG AGA TGT GTA TAA GAG ACA GGA CTA CHV GGG TAT CTA ATC C-3′ and KAPA HiFi HotStart ReadyMix enzyme (Roche, Pleasanton, CA, USA). This creates amplicons with protruding adapter strings that are compatible with the Illumina index and sequencing adapters. After preparation, the libraries were sequenced on a MiSeq system with chemistry v2 500 cycles. Sequencing was performed at the Nucleic Acid Sequencing New Generation Platform–RPT01J of *Fundação Oswaldo Cruz* (Rio de Janeiro, Brazil).

The initial sequence quality check was performed with FastQC [70]. Sequences were quality-filtered and trimmed using Trimmomatic version 0.36 [71], truncating reads if the quality dropped below 20 in a sliding window of 4 bp. USEARCH version 11.0.667 [72] was used for further processing, in order to merge and quality-filter sequencing reads, excluding reads with <350 or >470 nucleotides, in addition to reads with more than one ambiguous base or an expected error of more than 1. Filtered sequences were denoised and clustered into unique sequences (Amplicon Sequence Variants, ASV) using the UNOISE3 algorithm [73] implemented in USEARCH. Chimeric sequences were removed de novo during clustering and subsequently in reference mode using UCHIME with the Genome Taxonomy Database (GTDB, https://gtdb.ecogenomic.org (accessed on 1 May 2021) [74]. ASVs were classified against GTDB using the BLCA algorithm [75]. Sequences from mitochondria and chloroplast were removed from the dataset based in Greengenes 13_5 taxonomy [76]. The ASV table generated was then processed using the MicrobiomeAnalyst online software suite (https://www.microbiomeanalyst.ca/ (accessed on 1 May 2021)) in Marker Data Profiling mode. All the sequences that were generated in this study were deposited as a Sequence Read Archive in NCBI database.

Statistical Analyses. All data were analyzed using GraphPad Prism 6 (Graph-Pad Software, Inc., San Diego, CA, USA). Estimation of sufficient sample size was performed using the G* Power software (Version 3.1.9.2). A significance level of 5% and a power of 80% were considered for sample size estimation. Data are expressed as mean ± standard error of the mean (SEM). Statistical analyses were performed using the Student’s *t*-test, where the confidence interval was 95%.

## Figures and Tables

**Figure 1 ijms-23-05583-f001:**
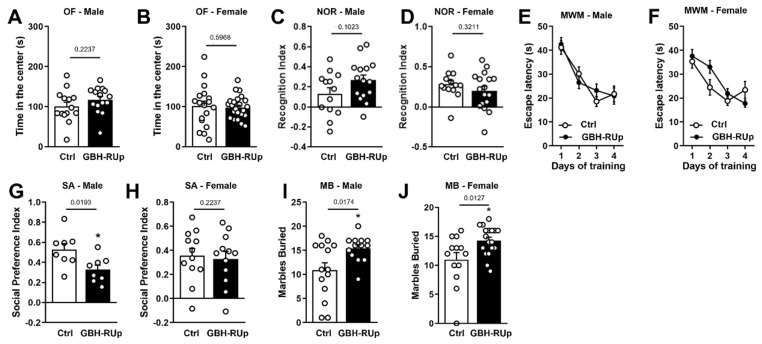
Long-term GBH-RUp exposure leads to behavioral changes in mice. Swiss mice were mated, and mice were exposed to GBH-RUp in drinking water through pregnancy, lactation, weaning, and into adulthood. When litters reached 3 months of age, they were submitted to several behavioral tasks. (**A**,**B**) Time spent in the center of the open field (OF) apparatus by male (**A**) and female (**B**) mice exposed to glyphosate-based herbicide RoundUp (GBH-RUp) or control mice (Ctrl), exposed to regular drinking water. (**C**,**D**) Recognition index in the novel object recognition task (NOR) for male (**C**) and female mice (**D**) exposed to GBH-RUp or Ctrl. (**E**,**F**) Latency to find the submerged platform in the Morris Water Maze (MWM) task by male (**E**) and female (**F**) mice exposed to GBH-RUp or Ctrl. (**G**,**H**) Social preference index for male (**G**) and female (**H**) mice exposed to GBH-RUp or Ctrl in the social approach (SA) task. (**I**,**J**) Number of marbles buried by male (**I**) and female (**J**) mice exposed to GBH-RUp or Ctrl, in the marble burying (MB) task. Symbols represent individual animals. In (**E**) and (**F**), data were analyzed by two-way ANOVA followed by the Bonferroni test. * *p* < 0.05 in Student’s *t* tests.

**Figure 2 ijms-23-05583-f002:**
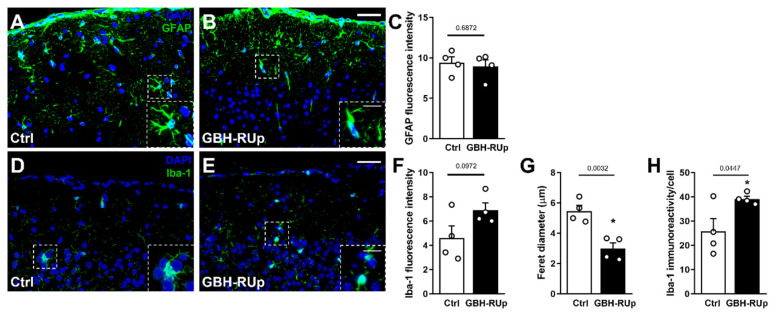
Long-term GBH-Rup exposure is associated with morphological changes in brain resident glial cells. Swiss mice were mated, and mice were exposed to GBH-RUp in drinking water through pregnancy, lactation, weaning, and into adulthood. After behavioral tests, mice were sacrificed, and brain samples were collected and processed for morphological analysis. (**A**,**B**) Representative images of GFAP immunostaining in the cortex of Ctrl (**A**) and GBH-RUp-exposed mice (**B**). Graph represents the quantification of GFAP fluorescence intensity (**C**) in the cortex of Ctrl and GBH-RUp-exposed mice. (**D**,**E**) Representative images of Iba-1 immunostaining in the cortex of Ctrl and GBH-RUp-exposed mice €. (**F**–**H**) Graphs represent the quantification of Iba-1 fluorescence intensity (**F**), Feret diameter of Iba-1 positive cells (**G**), and Iba-1 immunoreactivity per cell (**H**) in the cortex of Ctrl and GBH-RUp-exposed mice. Scale bar main images = 50 µm. Scale bar insets = 20 µm. Symbols represent individual animals. * *p* < 0.05, Student’s *t* tests.

**Figure 3 ijms-23-05583-f003:**
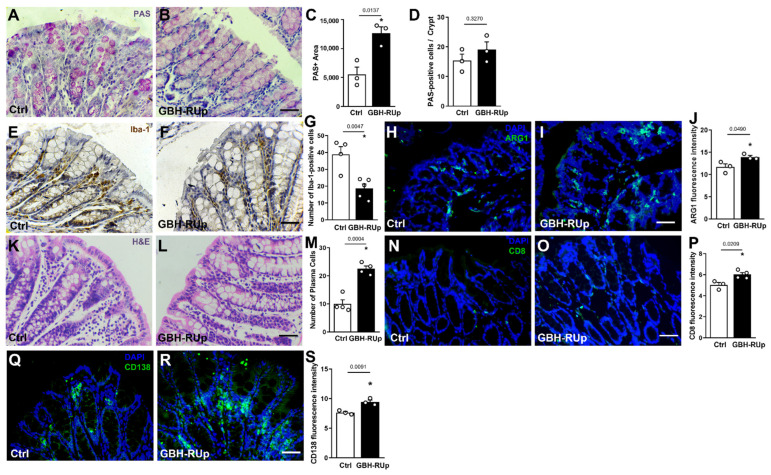
Long-term GBH-RUp exposure leads to abnormal cell distribution and increased infiltration of leukocytes in colonic lamina propria of mice. Swiss mice were mated, and mice were exposed to GBH-RUp in drinking water through pregnancy, lactation, weaning, and into adulthood. After behavioral tests, mice were sacrificed, and colon samples were collected and processed for morphological analysis. (**A**,**B**) Representative images of PAS staining in the colon of Ctrl (**A**) and GBH-RUp-exposed mice (**B**). Graphs represent the quantification of PAS-positive area (**C**) and the number of PAS-positive cells per crypt (**D**) measured in the colon of Ctrl and GBH-RUp-exposed mice. (**E**,**F**) Representative images of Iba-1 immunostaining in the colon of Ctrl (**E**) and GBH-RUp-exposed mice (**F**). (**G**) Graph represents the quantification of Iba-1-positive cells in the colon of Ctrl and GBH-RUp-exposed mice. (**H**,**I**) Representative images of Arginase-1 (ARG-1) immunostaining in the colon of Ctrl (**H**) and GBH-RUp-exposed mice (**I**). (**J**) Graph shows the quantification of ARG-1 fluorescence intensity in the colon of Ctrl and GBH-RUp-exposed mice. (**K**,**L**) Representative images of Hematoxilin & Eosin staining in the colon of Ctrl (**K**) and GBH-RUp-exposed mice (**L**). (**M**) Graph represents the quantification of cells that are morphologically compatible with plasma cells in the colon of Ctrl and GBH-RUp-exposed mice. (**N**,**O**) Representative images of CD8 immunostaining in the colon of Ctrl (**N**) and GBH-RUp-exposed mice (**O**). (**P**) Graph represents the quantification of CD8 immunofluorescence in the colon of Ctrl and GBH-RUp-exposed mice. (**Q**,**R**) Representative images of CD138 immunostaining in the colon of Ctrl (**Q**) and GBH-RUp-exposed mice (**R**). (**S**) Graph represents the quantification of CD138 immunofluorescence in the colon of Ctrl and GBH-RUp-exposed mice. Scale bars = 50 µm. Symbols represent individual animals. * *p* < 0.05, Student’s *t* tests.

**Figure 4 ijms-23-05583-f004:**
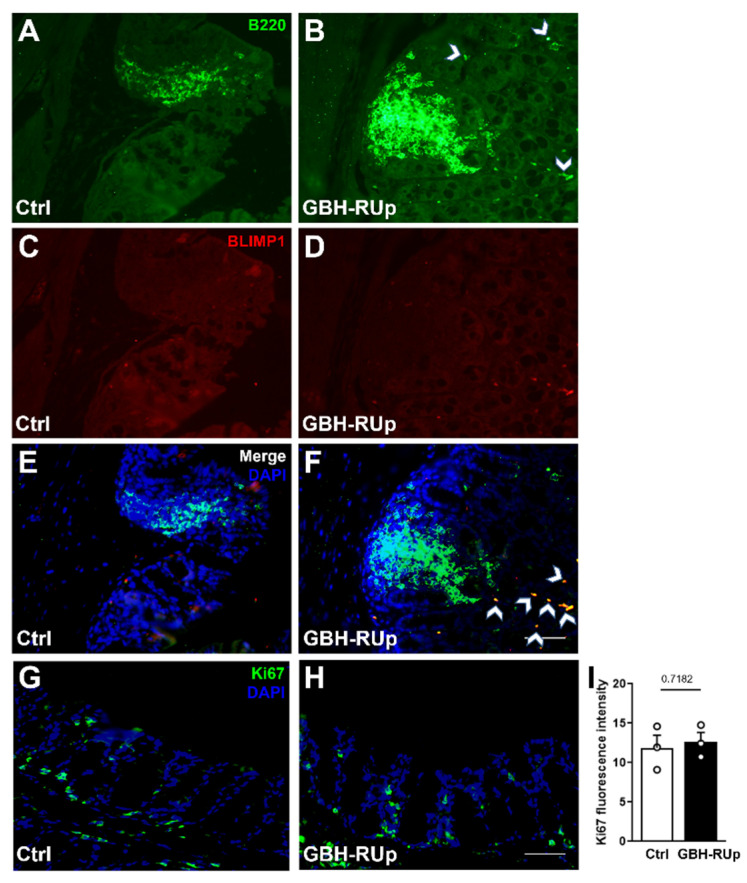
Long-term GBH-RUp exposure affect B cells niches in colonic lamina propria, interfering with lymphocytes differentiation into plasma cells. Swiss mice were mated, and mice were exposed to GBH-RUp in drinking water through pregnancy, lactation, weaning, and into adulthood. After behavioral tests, mice were sacrificed, and colon samples were collected and processed for immunofluorescence. (**A**,**B**) Representative images of B220 immunostaining in the colon of Ctrl (**A**) and GBH-RUp-exposed mice (**B**). Arrowheads indicate B220^+^ cells that are spread throughout the lamina propria. (**C**,**D**) Representative images of BLIMP1 immunostaining in the colon of Ctrl (**C**) and GBH-RUp-exposed mice (**D**). (**E**,**F**) Merge of B220 and BLIMP1 immunostaining in the colon of Ctrl (**E**) and GBH-RUp-exposed mice (**F**). Arrowheads indicate B220^+^BLIMP1^+^ cells that are spread throughout the lamina propria. (**G**,**H**) Representative images of Ki67 immunostaining in the colon of Ctrl (**G**) and GBH-RUp-exposed mice (**H**). (**I**) Graph represents the quantification of Ki67 immunofluorescence in the colon of Ctrl and GBH-RUp-exposed mice. Scale bars = 50 µm. Symbols represent individual animals. Data were analyzed by the Student’s *t* tests.

**Figure 5 ijms-23-05583-f005:**
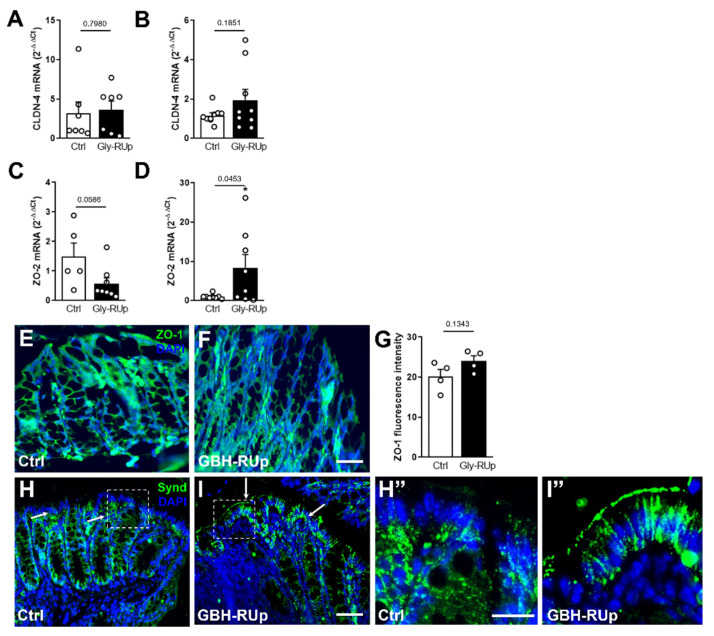
Long-term GBH-RUp exposure affects adhesion proteins’ expression, and leads to the redistribution of syndecan-1. Swiss mice were mated, and mice were exposed to GBH-RUp in drinking water through pregnancy, lactation, weaning, and into adulthood. After behavioral tests, mice were sacrificed, and colon samples were collected and processed for immunofluorescence and qPCR. (**A**,**B**) Graphs show the mRNA expression of *claudin-4* (*CLDN-4*) in the colon of 30 (**A**) or 60 (**B**) day-old Ctrl and GBH-RUp-exposed mice. (**C**,**D**) Graphs show mRNA expression of *zona occludens 2* (*ZO-2*) in the colon of 30 (**A**) or 60 (**B**) day-old Ctrl and GBH-RUp-exposed mice. (**E**,**F**) Representative images of zona occludens 1 (ZO-1) immunostaining in the colon of Ctrl (**E**) and GBH-RUp-exposed mice (**F**). (**G**) Graph shows the ZO-1 immunofluorescence intensity in the colon of Ctrl and GBH-RUp-exposed mice. (**H**,**I**) Representative images of syndecan-1 (Synd) immunostaining in the colon of Ctrl and GBH-RUp-exposed mice. Arrows indicate the distinct distribution of this protein in GBH-RUp-exposed mice. (**H”**,**I”**) Larger magnification images of indicated sections of panels H and I, respectively. Scale bars in (**E**–**I**) = 50 µm. Scale bars in (**H”**) and (**I”**) = 20 µm. Symbols represent individual animals. * *p* < 0.05, Student’s *t* tests.

**Figure 6 ijms-23-05583-f006:**
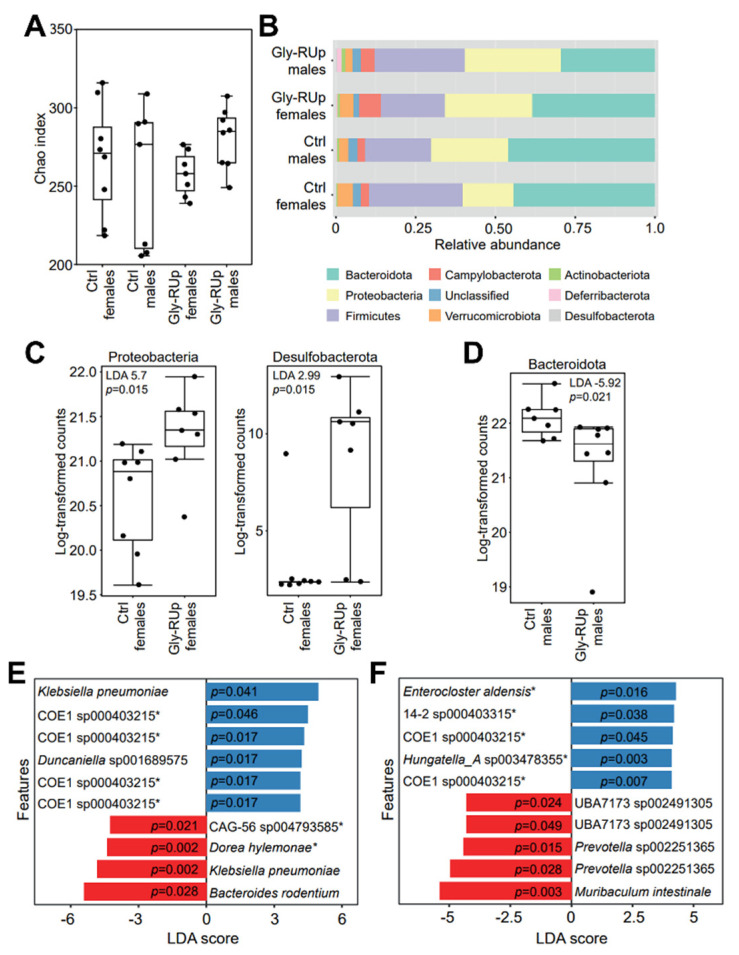
Effect of GBH-RUp treatment on the murine gut microbiota. (**A**) Alpha diversity (Chao index) of gut microbial communities from animals raised with or without GBH-RUp treatment. Symbols represent individual animals. (**B**) Bacterial composition, at the phylum level, of the gut microbiota of animals raised with or without GBH-RUp treatment. Data that are shown represent the averages of results from all mice. (**C**,**D**) LEfSe analysis results showing abundance levels for phyla with LDA scores >2 and *p* < 0.05 in female (**C**) and male (**D**) mice. Symbols represent individual animals. (**E**,**F**) Top 10 discriminating ASVs when comparing female (**E**) and male (**F**) animals with or without GBH-RUp treatment, as determined by LEfSe (LDA > 2, *p* < 0.05). Blue indicates taxa whose abundance was higher in Round Up-treated animals as compared to control animals, whereas red indicates taxa whose abundance was lower in GBH-RUp-treated animals. Asterisks indicate taxa within the Lachnospiraceae family. * *p* < 0.05.

**Table 1 ijms-23-05583-t001:** Antibodies used for immunohistochemistry and immunofluorescence.

Primary Antibody	Distributor	Secondary
Iba-1	Wako	Rabbit, Biotinylated
Arginase-1	Santa Cruz	Rabbit, Alexa Fluor-488
ZO-1	Invitrogen	Rabbit, Alexa Fluor-488
Ki67	Abcam	Rabbit, Alexa Fluor-488
Syndecan-1/CD138	BD Bioscience	Rat, Alexa Fluor-488
CD8	Abcam	Rabbit, Alexa Fluor-488
B220	BD Bioscience	Rat, Alexa Fluor-488
Blimp-1	BD Bioscience	Alexa Fluor-546 conjugated
GFAP	Invitrogen	Rabbit Alexa Fluor-488

**Table 2 ijms-23-05583-t002:** Primer sequences.

Target Gene		Primer Design	Product Size
β-actin	Forward	GCC CTG AGG CTC TTT TCC AG	51
	Reverse	TGC CAC AGG ATT CCA TAC CC	
ZO-2	Forward	GCC TGC AAG AAG GAG ACC AG	132
	Reverse	CTC GGC TCT GAG CCA AAA TG	
ZO-3	Forward	GTG TCG TGA GCT TCC CCA AG	156
	Reverse	ATG GCA TAC CAT TCA CCT GCA	
CLDN-4	Forward	TCG TGG GTG CTC TGG GGA T	170
	Reverse	GCG GAT GAC GTT GTG AGC G	

## Data Availability

Not applicable.

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
