# Peer review of "Lifelong Exposure to a Low-Dose of the Glyphosate-Based Herbicide RoundUp® Causes Intestinal Damage, Gut Dysbiosis, and Behavioral Changes in Mice"

_ijms, 2022, doi:10.3390/ijms23105583_

Round 1

Reviewer 1 Report

This is an interesting manuscript examining the effects of a glyphosate-based herbidice in mice, focusing on their gut and on their behavior.

Here follows a list of problems I have to raise.

1) Gly is the standard shortening for the aminoacid glycine. Please choose a different shortening for glyphosate.

2) A major problem with this study is that you tested a life-long exposure and this cannot be correlated to the development of a neurodevelopmental disorder such as autism; in fact, by definition, the causative factors of a neurodevelopmental disorder must act during the neurodevelopment and therefore you should have tested the effect either after an exposure in a time frame mimicking the first 1000 days of life in humans or in offspring of exposed mices.

3) A "Methods" section is supposed to come before the results, not at the end of the manuscript.

4) The "Results" section is poorly structured, as it contains methodological aspects, data analysis plan, but also data from existing literature, as well as the true results.

5) The behavioral examination of the mice needs to be supported by references.

6) Saying that there is an increase in a behavior means nearly nothing, unless this has been documented to be significant(i.e. the level of that behavior has been shown to be significant in indicating a behavioral alteration).

7)  I can find no data analysis plan and no statistical analysis in reported. Presenting only figures is not sufficient.

8) The Abstract is not sufficiently informative and should be rewritten, together with the whole manuscript.

8) Was a priori sample size determined? How? Or why not?

Reviewer 2 Report

This is an interesting study investigating the effects from lifelong exposure to low-dose of glyphosate-based herbicide in mice. The authors found that the herbicide causes intestinal damage, gut dysbiosis and autistic-like behavior. Overall, the study was well-organized and crucial experiments were conducted. The manuscript was also written in good English and overall was easy to understand and follow. One minor comment that I would recommend is to discuss the potential underlying mechanism that may cause the pathological and behavioral changes in these mice with lifelong exposure to the herbicide. Given the identification of a decrease/increase in several genes/proteins such as claudin-4, ZO-2, CD38, etc, please speculate (in the discussion) the potential underlying mechanisms that could drive these pathological and behavioral changes? 

Round 2

Reviewer 1 Report

The authors did a significant work on their manuscript to improve it. I appreciate the description of the methodology, which is now understandable. Still, I can see that there are a number of sentences referring to ASD (starting from the title... "autistic-like behaviors") that can be confusing to the reader.

Author Response

We thank the reviewer for his/her comment. We agree that we can not be sure that animals developed ASD-like behavior. Therefore, we have followed the reviewers suggestion by carefully revising the text to ensure that the behavioral results obtained in our study are not referred to as "autism" or "autistic-like behavior". All alterations in this revised version were performed using the "tracked changes" option in MS Word. We believe that this version is more transparent and will less confusing to readers.  

Round 3

Reviewer 1 Report

The authors effectively answered to my previous concerns.